# Enhancing Large Language Models through Adaptive Tokenizers

**Mengyu Zheng**
The University of Sydney
Huawei Noah's Ark Lab
mzhe4259@uni.sydney.edu.au

**Hanting Chen**
Huawei Noah's Ark Lab
chenhanting@huawei.com

**Tianyu Guo**
Huawei Noah's Ark Lab
tianyu.guo@huawei.com

**Chong Zhu**
Huawei Noah's Ark Lab
zhuchong4@huawei.com

**Binfan Zheng**
Huawei GTS AI Computing LAB
zhengbinfan1@huawei.com

**Chang Xu**
The University of Sydney
c.xu@sydney.edu.au

**Yunhe Wang**[*]
Huawei Noah's Ark Lab
yunhe.wang@huawei.com

## Abstract

Tokenizers serve as crucial interfaces between models and linguistic data, substantially influencing the efficacy and precision of large language models (LLMs). Traditional tokenization methods often rely on static frequency-based statistics and are not inherently synchronized with LLM architectures, which may limit model performance. In this study, we propose a simple but effective method to learn tokenizers specifically engineered for seamless integration with LLMs. Initiating with a broad initial vocabulary, we refine our tokenizer by monitoring changes in the model's perplexity during training, allowing for the selection of a tokenizer that is closely aligned with the model's evolving dynamics. Through iterative refinement, we develop an optimized tokenizer. Our empirical evaluations demonstrate that this adaptive approach significantly enhances accuracy compared to conventional methods, maintaining comparable vocabulary sizes and affirming its potential to improve LLM functionality.

## 1 Introduction

In recent years, large language models (LLMs) have emerged as foundational tools across a spectrum of applications in natural language processing [3, 7, 24]. From generating human-like text to enabling complex question-answering systems [28], LLMs have proven to be exceptionally versatile and capable. At the core of these models lies the tokenizer, a critical component that dictates how natural language is transformed into a format amenable to computational processing. The effectiveness of a tokenizer directly influences the model's ability to understand and generate language, thus playing a pivotal role in the overall performance of the LLM. Recognizing this integral relationship, it becomes essential to develop tokenizers that are not only effective but also dynamically adaptable to the evolving architectures of contemporary LLMs.

Current tokenization methods for large language models (LLMs) primarily include Byte Pair Encoding (BPE) [32], WordPiece [43], and Unigram [19], each serving to enhance text preprocessing by splitting

---

[*]Corresponding author

38th Conference on Neural Information Processing Systems (NeurIPS 2024).

it into manageable subwords. BPE focuses on reducing the dataset size through a greedy merging strategy based on character or subword frequency, effectively addressing the issue of infrequent words by splitting them into more common subunits. WordPiece, similar to BPE, starts with a base vocabulary and iteratively refines it by merging the most frequent pairs but incorporates a likelihood maximization step, which makes it slightly more context-aware than BPE. Unigram tokenization operates somewhat inversely, beginning with a large vocabulary and iteratively pruning it down based on token utility calculated through negative log likelihood, aiming to optimize the vocabulary against corpus loss metrics. Despite their efficiency in handling large vocabularies and improving computational feasibility, these tokenization methods are typically fixed once developed and are not designed to adapt or learn from the model's evolving understanding of language during training.

While traditional tokenization methods have been instrumental in enhancing the efficiency and effectiveness of large language models (LLMs), they are typically decoupled from the model's learning mechanisms. This means they do not adapt or evolve based on the model's performance or the specific requirements of the tasks being addressed. Instead, these methods prioritize compressing the vocabulary size, which can sometimes lead to suboptimal performance in complex language tasks where adaptability and contextual understanding are crucial. Recent advances in end-to-end learnable tokenization [17, 16, 38] aim to address these deficiencies by more closely integrating tokenization with the model's learning processes. However, these systems, while innovative, introduce significant computational overhead (*e.g.*, gradient-based tokenization and pooling modules) and lack the flexibility of traditional tokenization methods, which can be easily transferred across different models.

In this study, we address the limitations of traditional tokenization methods by creating a system where the tokenizer's development is coupled with the performance of the LLM itself. Specifically, we introduce an adaptive tokenizer that begins with a comprehensive initial vocabulary. As training progresses, we fine-tune this tokenizer by closely monitoring the model's perplexity. This ongoing adjustment allows the tokenizer to evolve in tandem with the LLM, ensuring that the tokenization process remains optimally aligned with the model's dynamic learning patterns. Our empirical results confirm that this adaptive approach markedly improves accuracy over traditional methods, demonstrating its potential to significantly enhance LLM functionality.

## 2 Related Works

### 2.1 Subword tokenizer

Subword tokenizers are widely applied in many large language models (LLMs) [32, 19, 42, 35, 20], such as GPT-3 [7], BERT [12], and T5 [30]. This is because subword tokenizers do not face the out-of-vocabulary issues that word-level tokenizers do. Unlike character-level tokenizers [36, 27], they do not require processing longer sequences at the character level, which significantly increases the complexity of the models quadratically [21]. Specifically, BPE (Byte Pair Encoding) [32] applies a compression algorithm [13] to the task of word segmentation. Unlike BPE, which builds a vocabulary from smaller to larger units, Unigram [19] starts by preparing a large seed vocabulary. The vocabulary is pruned until it reaches the specified size. Similar to the Unigram framework and its assumptions, BytePiece [35] trims the vocabulary based on token frequency to the required size. In its initial text encoding phase, BytePiece converts the corpus into bytes, enhancing the training speed of the tokenizer and achieving a higher compression rate. Obviously, these tokenizers are data-driven [5], with the generated vocabulary built based on the frequencies of word fragments [6]. However, they cannot directly interact with LLMs to enhance model capabilities [9].

### 2.2 Learnable tokenizer

Unlike traditional tokenizers that offer a predefined and fixed vocabulary, learnable tokenizers [17, 16, 38, 37, 34, 8] can be integrated with large language models into an end-to-end learning framework, resulting in task-specific tokenization to enhance the performance of LLMs. MANTa [16] introduces a gradient-based tokenization and pooling module that can be jointly learned with an encoder-decoder LLM [2]. RETVec [8] embeds words into a high-dimensional vector with a pre-trained model to be robust against adversarial attacks. Neural [17] adapts the tokenization behavior to the downstream task after pre-training the tokenization by distilling from a language-specific subword tokenizer. However, such tokenizers have high requirements for the quantity and quality of the training data. If

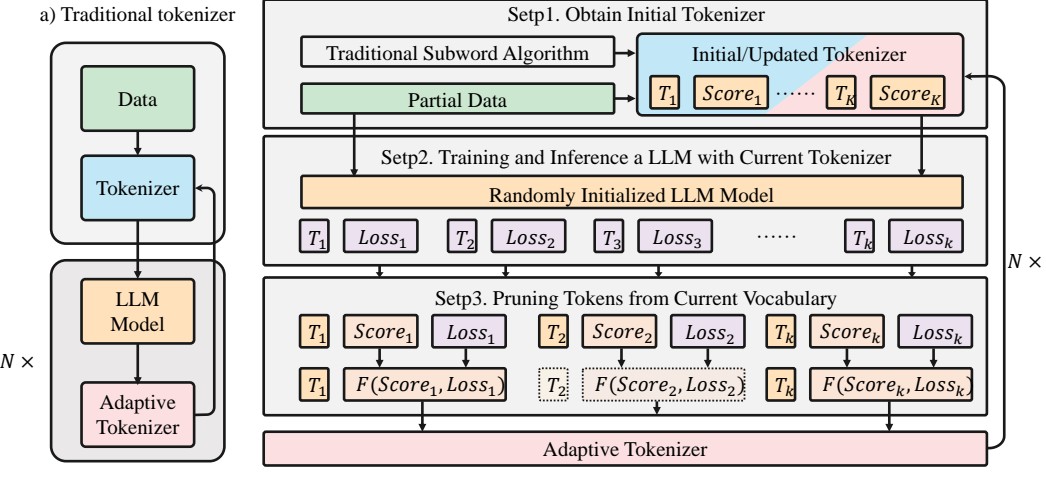

a) Traditional tokenizer

b) LLM enhanced tokenizer

c) Pipeline of the proposed ADAT

Figure 1: Illustration of the proposed ADAT pipeline. (a) The traditional tokenizer algorithm that directly extracts vocabulary from data. (b) The framework of the LLM-enhanced tokenizer, iteratively refining vocabulary based on model feedback. (c) Overview of ADAT, encompassing initial tokenizer acquisition, training and inference to derive token losses, token pruning based on scores and losses.

the data distribution is unbalanced or contains too much noisy data, it can lead to poor generalization of the tokenizer and negatively affect the performance of LLMs. For instance, Neural [17] requires a pre-training dataset curated with space-separated tokens and two carefully crafted heuristics to improve the ground label of the dataset. Therefore, stringent requirements for the quality of training data limit their widespread application.

In addition, some existing works have applied the concept of adaptive tokenizers in several fields, such as neural machine translation [26], domain adaptation [31], and text generation [23]. These task-adaptive tokenizers integrate tokenizers generated from different data distributions, focusing on how to combine the task-specific tokenizer with the other one. In contrast, our proposed model, ADAT, is designed to learn a general tokenizer. Therefore, the purpose of ADAT is different from that of the aforementioned adaptive tokenizers.

## 3 Adaptive Tokenizers

For Large Language Models (LLMs), two critical aspects are accuracy and inference speed, both of which are deeply intertwined with the design of the tokenizer. Specifically, an optimal vocabulary $V$ with maximum size $N$ in an objective dataset $D$ can be described by the following optimization problem:

$$\min_{V} \textbf{Length}(D_o, V) - \lambda \textbf{Acc}(D, M, V), \quad |V| \leq N, \tag{1}$$

where $M$ denotes the trained LLM, **Length** and **Acc** denotes the sequence length and accuracy (or the performance) using vocabulary $V$ in dataset $D$, and the $\lambda$ is a hyper-parameter to balance the two terms. This problem incorporates two primary objectives: given a fixed vocabulary size, the first is to maximize inference speed given a fixed vocabulary size, and the second is to maximize model accuracy. However, existing tokenizer schemes typically focus on one aspect over the other; for instance, traditional frequency-based schemes emphasize speed, while end-to-end approaches prioritize accuracy. To address this, we propose a method that optimizes both aspects. Therefore, we introduce an improved approach based on traditional frequency statistics, termed "adaptive tokenizers." This method aims to refine the balance between vocabulary efficiency and performance, thereby enhancing both the speed and accuracy of the model.

### 3.1 Unigram Model

Traditional tokenization methods generally fall into two categories: one approach, exemplified by Byte Pair Encoding (BPE) and WordPiece, starts with a small set of symbols and incrementally builds a larger vocabulary by merging the most frequent adjacent pairs. The other approach, typified by the Unigram method, begins with a large initial vocabulary which is progressively pruned based on token utility, a method found to generally offer superior performance due to its probabilistic foundation.

The Unigram model operates on the principle that the probability of a sentence is determined by the individual probabilities of its tokens. Here we briefly review the Unigram model. Initially, a large vocabulary $V$ is established. This extensive initial set includes potentially every unique word or subword unit observed in the training corpus, ensuring that the vocabulary can cover all possible textual inputs. The process of refining the vocabulary involves several key steps repeated in cycles: 1. **Probability Estimation**: For each token $x_i$ in the current vocabulary, we calculate its probability $p(x_i)$ based on its frequency of occurrence in the corpus. 2. **Loss Calculation**: We then compute the loss for each token, which is determined by how much the overall loss of the model would decrease if that token were removed. The loss function is calculated as:

$$\mathcal{L}_P(V) = \sum_{s=1}^{|V|} \log(p(X^{(s)})) = \sum_{i=1}^{|V|} \log(\sum_{\mathbf{x} \in S((X^{(s)})}^{P} (\mathbf{x})), \tag{2}$$

where $S(X)$ is a set of segmentation candidates built from the input sentence $X$, and $\mathbf{x} = (x_1, ..., x_K)$ is a subword sequence that $P(\mathbf{x}) = \prod_{i=1}^{K} p(x_i)$. The loss for each token $x_i$ is then formulated as $\mathcal{L}_P(x_i) = \mathcal{L}_P(V) - \mathcal{L}_P(V - x_i)$. 3. **Token Pruning**: Tokens are ranked according to their calculated loss. Finally, a proportion of tokens contributing the most to increasing the overall loss is pruned from the vocabulary.

### 3.2 LLM-Enhanced Tokenization

To enhance the integration of Large Language Models (LLMs) with our tokenization process, we have developed a simple but effective refined method for calculating the loss associated with each token, incorporating insights directly from the LLM's performance metrics. This approach aims to optimize the tokenizer's vocabulary to better align with the LLM's understanding and generation of text. The framework is illustrated in Figure 1.

Our revised loss calculation method integrates the traditional Unigram model's frequency-based loss with a performance-driven loss derived from an LLM. Specifically, we first train an LLM $M$ in the vocabulary $D$ using a training dataset $T$. This model is designed to capture the linguistic nuances relevant to the tasks it is trained for, providing a robust framework for assessing token utility. For each token $x_i$ in the vocabulary, we measure its individual contribution to the model's error using a cross-entropy loss function. The loss for each token is calculated as:

$$\mathcal{L}_M(x_i) = \sum_{x_i \in T} CE(M(x_{i-1}), x_i). \tag{3}$$

Here, $CE$ represents the cross-entropy function, $M(x_{i-1})$ is the LLM's output given the previous token $x_{i-1}$, and $x_i$ is the actual next token. This formula assesses how well the LLM predicts each token following its predecessor, providing a direct measure of each token's impact on model performance. Finally, The cross-entropy loss for each token is then combined with the traditional Unigram frequency-based loss. This combined loss ensures that tokens are evaluated not only on their frequency of occurrence but also on their actual contribution to the LLM's task performance. The final loss for pruning the vocabulary is given by:

$$\mathcal{L}(x_i) = F(\mathcal{L}_P(x_i), \mathcal{L}_M(x_i)), \tag{4}$$

where $F(\cdot, \cdot)$ is a function to balance the importance of frequency-based loss and LLM-driven loss, which will be discussed in experiments. Using this enhanced loss metric, we iteratively refine the vocabulary by pruning tokens that contribute the least to the combined loss, thus optimizing the vocabulary for both general language understanding and specific task performance. This process continues until the vocabulary is compact enough to manage while still being comprehensive enough to support the LLM effectively.

**Random sampling.** In the training of Large Language Models (LLMs), ensuring that each token within the set vocabulary receives equal and substantial training is crucial to prevent loss bias due to uneven training. While iterating over all possible tokenizations of the training corpus would ideally provide the most comprehensive learning experience, this approach is computationally prohibitive due to the immense variety of potential segmentations. To address this, we adopt the classic Viterbi algorithm [40] to perform randomized tokenization of the training data. This method allows for a diverse and balanced exposure of all tokens within the vocabulary to the learning process. The Viterbi algorithm efficiently determines the most probable tokenization paths through a probabilistic model of token occurrence, which significantly reduces the computational overhead compared to exhaustive methods. By leveraging this approach, our LLM can learn each token in the vocabulary more uniformly, enhancing the overall robustness and performance of the model.

**Loss momentum.** In the iterative process of training Large Language Models, maintaining the stability of the vocabulary is crucial to ensure consistent learning outcomes. To achieve this, we propose a momentum-based improvement for calculating the loss during each iteration. Specifically, the loss for iteration $j$ of token $x_i$, denoted as $\mathcal{L}^j(x_i)$, is not solely computed based on the current data but is also weighted by the loss from the previous iteration $L^{j-1}$. This approach allows for a smoother convergence and mitigates fluctuations in training dynamics. The formula for updating the loss at each iteration is given by:

$$\mathcal{L}^j_{\text{momentum}}(x_i) = \beta \mathcal{L}^{j-1}_{\text{momentum}}(x_i) + \mathcal{L}^j(x_i), \tag{5}$$

where $\beta$ is the momentum coefficient that controls the extent to which the previous loss influences the current loss. This methodology not only stabilizes the vocabulary updates across iterations but also enhances the model's ability to generalize from the training data by reducing the variability in loss across successive training epochs.

## 4 Experiments

In this section, we outline the comprehensive experimental framework designed to assess the effectiveness of our proposed tokenizer, **Ada**ptive **T**okenizer (**ADAT**), in comparison to established methods such as Byte Pair Encoding (BPE) [32] and the Unigram model [19]. These evaluations utilize the Pythia [3] suite of models at various scales, leveraging a substantial corpus to ensure robust and generalizable results.

### 4.1 Experimental Setup

**Model Framework**   We deploy the Pythia framework [3] for its lightweight design and adaptability across different computational setups. Pythia's flexibility facilitates reproducibility and consistent assessment of performance, making it an ideal choice for evaluating the scalability and efficiency of various tokenization strategies across model sizes of 70M, 160M, and 410M parameters.

**Data Corpus**   The study utilizes a substantial corpus extracted from The Pile [14], consisting of 56GB of raw data across 91 files. We specifically excluded subsets from DM_Mathematics and Github to ensure the relevance and quality of the data. The remaining data, approximately 16 billion tokens after a random shuffle, was tokenized using a Unigram [19] tokenizer with a vocabulary size of 50,000 tokens. A detailed enumeration of the data files used is available in Supp. A.8.

**Baseline Methods**   Our investigation compares four tokenization methods: Bytepiece [35], Byte Pair Encoding (BPE) [32], Unigram [19], and our proposed **ADAT**. These tokenizers were selected based on their established efficacy in handling large corpora and their theoretical implications for processing complex linguistic data.

**Evaluation Metrics**   The effectiveness of each tokenization strategy is rigorously evaluated using several metrics. These include Perplexity (PPL), which measures the model's predictive accuracy, and Compression Rate(refer to A.1), assessing how efficiently the tokenization process reduces vocabulary size while preserving linguistic diversity. We calculate PPL for all models on PG19 [29] dataset. Specifically, we use its test set and the first 2048 tokens for each book. Furthermore, we use the Language Model Evaluation Harness [15] to run five-shot evaluations on eight common language

Table 1: Performance Comparison of Different Tokenization Methods.

| Metric | BPE | BytePiece | +ADAT(Ours) | Unigram | +ADAT(Ours) |
|---|---|---|---|---|---|
| PPL | 22.31 | 71.5 | 67.19(-4.31) | 16.52 | **6.97(-9.55)** |
| ARC-C | $17.32 \pm 1.11$ | $18.69 \pm 1.14$ | $18.94 \pm 1.15$ | **19.54**$\pm1.16$ | $18.46 \pm 1.12$ |
| ARC-E | $37.58 \pm 0.99$ | $33.80 \pm 0.97$ | $33.71 \pm 0.97$ | $37.04\pm0.99$ | **40.57**$\pm0.99$ |
| Boolq | $61.28 \pm 0.85$ | $42.12 \pm 0.87$ | **62.20**$\pm0.85$ | $53.06 \pm 0.87$ | $61.19 \pm 0.85$ |
| Lambda | $10.89 \pm 0.43$ | $8.80 \pm 0.39$ | $13.55 \pm 0.48$ | $17.27 \pm 0.53$ | **17.97**$\pm0.52$ |
| LogiQA | $23.04 \pm 1.65$ | $20.28 \pm 1.58$ | $22.27 \pm 1.63$ | $23.20 \pm 1.66$ | **24.22**$\pm1.70$ |
| PIQA | $59.25 \pm 1.15$ | $57.83 \pm 1.15$ | $56.96 \pm 1.16$ | **60.50**$\pm1.14$ | $59.93 \pm 1.14$ |
| SciQ | $66.60 \pm 1.49$ | $54.01 \pm 1.58$ | $51.90 \pm 1.58$ | $68.10 \pm 1.47$ | **72.40**$\pm1.44$ |
| SST-2 | $51.26 \pm 1.69$ | $49.08 \pm 1.69$ | $50.23 \pm 1.69$ | $49.77 \pm 1.69$ | **54.24**$\pm1.69$ |
| Winogrande | $49.96 \pm 1.41$ | $50.31 \pm 1.41$ | $49.41 \pm 1.41$ | $51.46 \pm 1.40$ | **51.62**$\pm1.40$ |
| Avg. (%) | 41.91 | 37.21 | 39.91(+2.70) | 42.22 | **44.51**$(+$**2.29**$)$ |

modeling benchmarks: Lambada (OpenAI) [25], PIQA [4], WinoGrande [1], ARC-Easy [10], ARC-Challenge [10], SciQ [18], LogiQA [22], and SST-2 [33, 41], to provide a comprehensive insight into each method's capabilities.

By analyzing the impact of tokenization on model scalability and the influence of vocabulary size variations, this study aims to enhance our understanding of how tokenization strategies can optimize language models for efficiency and linguistic performance. The findings are expected to contribute significantly to the development of more robust and adaptable language processing tools, catering to a wide array of NLP applications. The Pythia models are trained using a corpus of 15B tokens, where training the 70M model consumes approximately 48 GPU hours with FlashAttention [11]. The models used for loss calculation require additional 2 GPU hours.

## 4.2 Tokenization Methods Evaluation

In this section, we examine the effects of different tokenization strategies on the training effectiveness. The core objective is to explore how variations in the vocabulary, induced by different tokenization methods, affect model training and performance.

The Pythia-70M model is selected due to its moderate size and efficiency, which help mitigate the complexities associated with larger model architectures. It is initialized with random weights and undergoes a single training epoch using pre-training data. This data is processed with vocabularies generated from 1/10th of the training corpus (approximately 1.5 billion tokens), each containing 50,000 tokens—a size consistent with the Pythia [3] setup.

Baseline tokenization methods including BPE, Unigram, and BytePiece, generate vocabularies consisting of 50,000 tokens directly from initial data that approximately one-tenth of the training corpus (about 1.5 billion tokens). In contrast, for the proposed ADAT method, initial vocabularies are generated using either BytePiece or Unigram with 150,000 tokens. These are then methodically refined down to 50,000 tokens over 5 iterative steps, matching the baseline vocabulary size. At each step, a randomly initialized model is trained on approximately 0.3 billion tokens from the initial dataset. Subsequently, the model performs inference on a subset of 0.1 billion tokens, during which token loss is calculated. This loss data, when combined with token frequency using the formula $\frac{a}{\lambda \log(b+1)}$, guides the vocabulary pruning process. An ablation study on the combination methods will be discussed in Section 4.6.4.

As illustrated in Table 1, our method achieves its best performance when initialized with the Unigram vocabulary, recording a score of 44.51. This score represents a considerable improvement of 2.29 points over the standard Unigram model and surpasses the BPE model by 2.6 points. Additionally, our approach shows a notable enhancement of 2.7 points when utilizing BytePiece as the initial vocabulary. Although the BytePiece vocabulary generally exhibits inferior baseline results, our method effectively elevates its performance, indicating robustness across both high-quality (Unigram) and lower-quality (BytePiece) vocabularies. These results not only affirm the efficacy of our method but also demonstrate its adaptability to different initial conditions, thereby validating its potential for broad adeptness on diverse vocab initialization.

Table 2: Evaluation on Different Scale Model Size.

| Metric | 70M | | 160M | | 410M | |
|---|---|---|---|---|---|---|
| | Unigram | ADAT | Unigram | ADAT | Unigram | ADAT |
| PPL | 16.52 | **6.97(-9.55)** | 13.97 | **6.19(-7.78)** | 10.92 | **5.78(-5.14)** |
| ARC-C | **19.54**±1.16 | 18.46 ± 1.12 | 18.69 ± 1.14 | **18.94**±1.15 | **20.82**±1.19 | 19.29 ± 1.21 |
| ARC-E | 37.04±0.99 | **40.57**±0.99 | 39.52 ± 1.00 | **42.87**±1.01 | 44.65 ± 1.02 | **46.69**±1.03 |
| Boolq | 53.06 ± 0.87 | **61.19**±0.85 | **58.56**±0.86 | 57.68 ± 0.86 | 54.80 ± 0.87 | **60.81**±0.87 |
| Lambda | 17.27 ± 0.53 | **17.97**±0.52 | 19.06 ± 0.55 | **25.02**±0.60 | 27.81 ± 0.62 | **28.94**±0.66 |
| LogiQA | 23.20 ± 1.66 | **24.22**±1.70 | **25.65**±1.71 | 25.04 ± 1.65 | 23.20 ± 1.66 | **24.32**±1.68 |
| PIQA | **60.50**±1.14 | 59.93 ± 1.14 | 60.83 ± 1.14 | **61.86**±1.14 | 63.38 ± 1.12 | **64.61**±1.11 |
| SciQ | 68.10 ± 1.47 | **72.40**±1.44 | 72.10 ± 1.42 | **79.60**±1.28 | 80.70 ± 1.25 | **83.50**±1.14 |
| SST-2 | 49.77 ± 1.69 | **54.24**±1.69 | 52.06 ± 1.69 | **52.78**±1.69 | 50.69 ± 1.69 | **54.71**±1.69 |
| Winogrande | 51.46 ± 1.40 | **51.62**±1.40 | 49.88 ± 1.41 | **50.69**±1.41 | 52.41 ± 1.40 | **52.93**±1.41 |
| **Avg** | 42.22 | **44.51** | 44.04 | **46.05** | 46.50 | **48.42** |

## 4.3 Scalability

We examine the scalability of a proposed tokenization method that tailors the vocabulary to model size, unlike the static Unigram method which maintains a consistent vocabulary across various model capacities. The scalability of the tokenization methods is tested using the Pythia framework configured at three different levels of computational complexity: 70M, 160M, and 410M parameters. For each model size, our method generates an optimized vocabulary specific to that configuration, allowing us to analyze how adjustments in vocabulary affect performance as model size increases. In contrast, the Unigram method employs a uniform 50,000-word vocabulary across all sizes, serving as a baseline. We gauge performance using Perplexity (PPL) and scores from benchmark datasets designed to assess the linguistic capabilities of each model under various conditions, providing insights into the efficiency and adaptability of the tokenization methods at scale. For training larger models, the same volume of data will lead to insufficient warm-up, potentially resulting in a slight decline in the accuracy of loss computations used for determining token priority. As a result, we increase the data volume for training the loss calculation model according to the size of model.

The results of this experimental framework, as presented in Table 2, indicate substantial performance variations across different model sizes employing varied tokenization strategies. Specifically, average performance scores across all evaluated metrics demonstrate consistent improvements with increases in model sizes: ADAT achieves a score of 44.51 in the 70M model, significantly surpassing Unigram's 42.22; 46.05 compared to 44.04 in the 160M model; and 48.32 versus 46.50 in the 410M model. These findings highlight the superior efficacy of ADAT in managing diverse model volumes compared to the more static approach of Unigram, which exhibits limited scalability with increasing model size. Remarkably, the performance of the 70M model using ADAT exceeded that of the Unigram on the 160M model by nearly 5%, illustrating the substantial enhancement and ability of our method to bridge a parameter gap of over double. Furthermore, the performance of our 160M model approaches that of the 410M model, emphasizing the robust adaptability of the ADAT method across varying computational scales.

Table 3: Cross-Model Adaptability of Vocabularies.

| Model Size | Unigram | 70M-Model Vocabulary | 410M-Model Vocabulary |
|---|---|---|---|
| 70M | 42.22 | 44.51 | 42.62 |
| 160M | 44.04 | 45.03 | 45.83 |
| 410M | 46.50 | 47.66 | 48.42 |

## 4.4 Cross-Model Adaptability

This experiment evaluates the adaptability of vocabularies generated by our proposed tokenization method across various configurations of the Pythia model, particularly assessing whether vocabularies optimized for one model size can effectively scale to others. We initially create vocabularies using the 70M and 410M configurations. These are then used to train models at both scales to evaluate performance in downstream tasks, allowing us to assess how vocabularies designed for a specific

Table 4: Impact of Vocabulary Size on Model Performance Across Different Model Sizes.

| Model Size | Vocabulary Size | Tokenization Method | Accuracy | Perplexity (PPL) |
|---|---|---|---|---|
| 70M | 50,000 | **Unigram** | 42.22 | 16.52 |
| | | **ADAT** | 44.51(+2.29) | 6.97 |
| | 30,000 | **Unigram** | 40.93 | 32.53 |
| | | **ADAT** | 43.33(+2.40) | 7.38 |
| 160M | 50,000 | **Unigram** | 44.04 | 13.97 |
| | | **ADAT** | 46.05(+2.01) | 6.19 |
| | 30,000 | **Unigram** | 43.08 | 15.21 |
| | | **ADAT** | 45.26(+2.14) | 6.11 |

size perform when applied to both smaller and larger models, thus examining their cross-model adaptability.

Table 3 illustrates the cross-model adaptability of vocabularies across different model sizes. By applying vocabularies derived from different model sizes to various models, we observe that vocabularies generated by the 70M and 410M models surpass the performance of the standard Unigram model. This indicates the adaptability of the ADAT vocabularies across different model sizes. Furthermore, we note that the vocabulary from the 410M model achieves only a marginal improvement of 0.4 when applied to the 70M model, significantly less than the 2.29 increase afforded by the 70M model's vocabulary. This suggests that the vocabularies selected by ADAT possess a strong capacity for targeted optimization, enabling the selection of tokenization strategies that are specifically tailored to the characteristics of different models.

## 4.5   Model and Vocabulary Size

The experiment aims to assess the impact of different tokenizer strategies on model performance across two vocabulary sizes, comparing a standard 50,000-token set with a reduced 30,000-token set. We utilize two configurations of the Pythia model—70M and 160M—to explore how vocabulary size influences model efficiency. Each model is tested using both the standard Unigram and our proposed tokenization method. This setup allows us to directly observe the effects of reduced vocabulary sizes on the performance dynamics, providing insights into how smaller vocabularies impact the computational efficiency and efficacy of language models.

The experimental results, as presented in Table 4, support the hypothesis that changes in vocabulary size can significantly affect model performance, with different impacts observed across varying model sizes. For large language models, it is common for models of vastly different sizes to utilize vocabularies of similar or identical sizes [3, 39]. This practice can lead to issues of performance or efficiency. Our method offers a more effective solution by tailoring tokenization strategies to the specific sizes of models, thereby mitigating these challenges. For the 70M model, ADAT achieved a notable improvement in accuracy from 42.22 to 44.51 (+2.29) and a substantial reduction in perplexity from 16.52 to 6.97 when using a 50,000-word vocabulary. Even with a reduced vocabulary of 30,000, ADAT enhances accuracy to 43.33 (+2.40) and decreases perplexity to 7.38, suggesting robustness against vocabulary size reduction. In contrast, the 160M model, which has a greater parameter capacity, also shows improvements with ADAT: accuracy increases from 44.04 to 46.05 (+2.01), and perplexity drops sharply from 13.97 to 6.19 for the 50,000 vocabulary size. With a 30,000 vocabulary, accuracy still increases to 45.26 (+2.14), and perplexity remains low at 6.11, underscoring that larger models not only handle vocabulary reductions well but also benefit significantly in terms of computational efficiency and model quality.

## 4.6   Ablation Study

This ablation study is structured into three distinct parts to explore how variations in the inference corpus size used for calculating token loss, initial vocabulary sizes, momentum strategy, and balance function $F(a, b)$ influence the efficacy of our proposed tokenization method on a 70M parameter model. More ablation results can be referred to in the supplementary.

Table 5: Ablation Studies Results.

| Infer Data Volume | | Initial Vocabulary Size | | Momentum | | Balance $F(a,b)$ | |
|---|---|---|---|---|---|---|---|
| Tokens | Acc. | Init Size | Acc. | Methods | Acc. | Methods | Acc. |
| 1M | 43.13 | 75k | 43.42 | Unigram | 42.20 | $a - \lambda b$ | 42.70 |
| 10M | 43.74 | 100k | 43.78 | ADAT+By | 44.51 | $log(a) - \lambda b$ | 43.23 |
| 100M | 44.51 | 150k | 44.51 | -w/o Mnt. | 43.16 | $a/\lambda \log(b+1)$ | 44.51 |

### 4.6.1 Corpus Size used in Loss Calculation

We conducted an experimental study to investigate the effects of varying corpus sizes on the accuracy of token loss calculations. The experiment assessed the performance of models trained on different sizes of inference data, specifically 1 million (1M), 10 million (10M), and 100 million (100M) tokens. The results are summarized in the table 5.

These results indicate a direct correlation between the volume of the corpus used during the loss calculation phase and the overall accuracy of token loss estimates. When smaller corpora are used, a significant number of tokens are absent, resulting in numerous instances where loss values cannot be computed. Furthermore, the precision of token loss estimations tends to decrease with smaller data sets. Even with just 1M tokens, there was a noticeable improvement over the baseline unigram vocabulary accuracy of 42.22. This enhancement became more pronounced with larger data volumes, reaching an increase of 44.51 in accuracy with 100M tokens.

### 4.6.2 Initial Vocabulary Size

This segment of our study assesses the effect of different initial vocabulary sizes on model performance. Adjusting the vocabulary from a baseline of 150,000 tokens to either 100,000 or 75,000 tokens, we explore the influence of vocabulary scale on training outcomes. The results, detailed in Table 5, illustrate the trade-offs associated with varying vocabulary sizes.

From the experiment, it is evident that models equipped with a larger initial vocabulary of 150,000 tokens tend to achieve lower Perplexity and higher Accuracy, indicating a robust ability to capture diverse linguistic nuances that significantly enhance performance. In contrast, reducing the vocabulary size to 75,000 tokens results in increased perplexity and decreased accuracy, highlighting a potential compromise in linguistic detail that adversely affects model functionality, especially in complex linguistic scenarios.

### 4.6.3 Momentum Strategy

This experiment evaluates the impact of incorporating a momentum strategy into our tokenization algorithm's vocabulary pruning process. The performance of vocabularies pruned under both the momentum and non-momentum conditions is directly compared in Table 5.

The results in Table 5 demonstrate that the momentum approach significantly enhances model accuracy, with a notable improvement from 43.16% to 44.51% in the ADAT method with Unigram initialization vocabulary when momentum is applied. Similarly, the Unigram method shows a baseline performance of 42.20% accuracy. These results confirm that integrating momentum allows for a more refined pruning process by effectively utilizing historical performance data to make more informed decisions, thereby preserving valuable linguistic features.

### 4.6.4 Balance Strategy

In this ablation study, we investigate various functions to balance token frequency and loss value in our tokenization algorithm. The primary objective is to adhere to the principle that tokens with higher frequency and lower loss should be assigned higher priority. Given the significant difference in their magnitudes, we explored subtraction and division methods. We evaluated three functions, detailed in Table 5. Here, $\lambda$ is a scaling factor introduced to adjust the balance between frequency and loss, and we set it as 1 in practice.

The results demonstrate that the subtraction methods $a - \lambda b$ and $a - \lambda \log(b)$ yielded accuracies of 42.70 and 43.23 respectively. These results indicate relatively poor performance, even with

the logarithmic transformation applied to the score $\log(a) - \lambda b$. This underperformance is likely attributable to the significant disparity in the magnitudes of frequency and loss values, which the subtraction methods struggle to reconcile effectively. In contrast, the division method $\frac{a}{\lambda \log(b+1)}$ significantly outperformed the subtraction approaches with an accuracy of 44.51. This superior performance suggests that the division method more naturally balances the influence of frequency and loss by scaling the loss logarithmically before the division, thereby mitigating the impact of numerical range discrepancies. This method's ability to integrate frequency and loss without requiring additional adjustments for scale disparities results in more stable and effective prioritization of tokens.

### 4.7 Analysis of the Compute Costs

To prove that the proposed method is feasible in practice. We analyzed the empirical runtime introduced by ADAT. To measure runtime, we used 8 NVIDIA A100 GPUs, an Intel 8378A CPU, and PyTorch 2.1.2 with CUDA 12.1. The tokenizer optimization involves 5 epochs, where each epoch consists of training the LLM on a 0.3B corpus, followed by inference on a 0.1B corpus, and concludes with a vocabulary pruning step (90 seconds for a 100K tokens vocabulary). Therefore, the total computational cost of the tokenizer optimization process is calculated as:

$$5 \times (0.3B \text{ training} + 0.1B \text{ inference} + \text{ pruning time}) = 1.5B \text{ training} + 0.5B \text{ inference} + 450s.$$

ADAT introduces an additional training cost of 1.5B tokens and an inference cost of 0.5B tokens, along with minimal vocabulary pruning time. Compared to the hundreds of billions or even trillions of tokens required for LLM training, these computational costs are negligible. As shown in Table 6, the full-scale training of the LLM incurs significantly higher computational costs. For instance, when training models with a 16B and 60B corpus, the tokenizer optimization accounts for only 4.17% and 1.04% of the total training time, respectively. The Pythia-70M model takes 510 GPU hours to train with the full Pile dataset [3], and exceeds the tokenizer optimization's computational cost by over 255 times. Therefore, the additional computational cost introduced by our method is minimal, making it feasible in practice.

Table 6: Runtime of ADAT optimization and training models.

|  | Tokenizer Optimization | Training on 16B | Training on 60B | Pythia Report |
|---|---|---|---|---|
| Runtime | 2 GPU hours | 48 GPU hours | 192 GPU hours | 510 GPU hours |

## 5 Limitations

The adaptive nature of our proposed tokenizer method introduces variations in tokenizers across different model sizes, leading to inconsistent vocabularies. This inconsistency complicates tasks such as knowledge distillation and speculative decoding, which rely on the assumption of a uniform vocabulary across both smaller and larger models.

## 6 Conclusion

In this paper, we have presented a novel approach to tokenizer design that integrates key aspects of both accuracy and inference speed, addressing the inherent limitations found in existing tokenizer schemes. By innovating beyond the traditional frequency-based and end-to-end methodologies, our adaptive tokenizer framework strategically optimizes vocabulary construction, ensuring both rapid processing and high precision in language modeling tasks. Our results demonstrate that the adaptive tokenizer significantly enhances the performance of Large Language Models (LLMs) across various benchmarks, providing a balanced solution that does not sacrifice speed for accuracy or vice versa. Future work will focus on refining these adaptive tokenization techniques, exploring further integration with neural network architectures, and expanding their applicability to a broader range of languages and complex linguistic tasks.

## Acknowledgements

This work was supported by the Australian Research Council under Projects DP240101848 and FT230100549.

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

# A  Appendix

## A.1  Compression Rate results.

The compression rate of the proposed model and baselines across Pythia model[2]-70M models are shown in Table 7.

Table 7: Performance comparison of different tokenization methods.

| Metric | BPE | BytePiece | ADAT+By | Unigram | ADAT+U |
|---|---|---|---|---|---|
| Compression Rate | 4.38 | 4.81 | 4.98 | 3.96 | 2.91 |

## A.2  Results of ADAT in the 1B model.

To further demonstrate the scalability of our proposed ADAT method, we expanded our experimental results on larger models and larger corpus. Specifically, we add results(shown in Table 8) from training a 1B model on 60B corpus. The results demonstrate that on larger models and with more data, ADAT continues to show substantial improvements over the baseline, indicating that ADAT has strong scalability.

Table 8: Results in the 1B model.

| Metric | Unigram | ADAT |
|---|---|---|
| Avg | 49.11 | 51.20 |

## A.3  Analysis of Differences Between Vocabularies

To illustrate the differences between the vocabulary results obtained by ADAT and those from unigram, we calculated the overlap ratio of the two token sets as follows:

$$Ratio = |A_{vocab} \cap B_{vocab}|/|A_{vocab} \cup B_{vocab}|. \tag{6}$$

Where $\cap$ and $\cup$ denote the intersection and union of two sets, respectively, we present the overlap ratios between the tokenizers obtained using ADAT and Unigram on different models under the same vocabulary size setting, as well as the overlap ratios between these tokenizers themselves in Table 9. As shown in Table 1, there are significant differences between the vocabulary generated by ADAT and that generated by unigram. This disparity arises because ADAT is not entirely data-driven in its vocabulary generation process. By incorporating the loss from the LLM, ADAT can simultaneously focus on enhancing the performance of the LLM. Additionally,the overlap ratio between ADAT-160M and ADAT-410M is higher than that between ADAT-160M and ADAT-70M. This also indirectly explains why, as shown in Table 3, the tokenizer generated by ADAT-410M is more suitable for Pythia-160M compared to the tokenizer generated by ADAT-70M.

Table 9: Comparison between Vocabulary of ADAT and Unigram.

| Unigram vs. | | ADAT | |
|---|---|---|---|
| Model size | Ratio | Model size | Ratio |
| ADAT-70M | 0.18 | 70M \| 160M | 0.71 |
| ADAT-160M | 0.19 | 70M \| 410M | 0.69 |
| ADAT-410M | 0.21 | 160M \| 410M | 0.84 |

To compare the vocabulary obtained by ADAT with the initial vocabulary, we sorted the 100k initial vocabulary by score in descending order and calculated the percentage of tokens, pruned by ADAT,

---

[2]with Apache-2.0 license

that fall into each score interval. As shown in Table 10, ADAT-70M-50K tokens are most densely distributed not in the 0-25% interval, indicating that ADAT relies not only on token frequency but also on the prediction difficulty of tokens in the LLM during training.

Table 10: Comparison with Initial Vocabulary with vocabulary size 100k.

| Vocabulary | 0-25% | 25-50% | 50-75% | 75-100% |
|---|---|---|---|---|
| Unigram-50k | 54.75% | 23.95% | 8.11% | 13.19% |
| ADAT-70M-50k | 21.43% | 31.60% | 32.72% | 14.24% |

## A.4   Impact of Training Epochs

We investigate the effects of varying the number of training epochs for developing the vocabulary. Specifically, the model is trained using vocabularies that have been developed over 3, 5, and 7 epochs. The model is initialized with random weights for each training session to evaluate the immediate impact of the epoch variation.

As shown in Table 11, the results from varying the number of training epochs suggest a clear relationship between training duration and vocabulary efficacy. Before the epoch reaches a certain value (5 epochs), increasing the number of epochs benefits accuracy, indicating a more refined and effective vocabulary. However, accuracy does not significantly improve with further increases in the number of epochs. This suggests that, given the specified vocabulary size, ADAT can quickly and efficiently learn an appropriate vocabulary without the need for prolonged training over many epochs. Therefore, in our experiments, the default number of training epochs is set to 5.

Table 11: Impact of Training Epochs on Model Performance.

| Training Epochs | Accuracy |
|---|---|
| 3 | 43.67 |
| 5 | 44.51 |
| 7 | 44.58 |

## A.5   Expanded Analysis on Infer Data Volume Tokens and Initial Vocabulary Size

we have expanded the analysis of 'infer data volume tokens' and 'initial vocabulary size'—both variables are explored within and beyond settings in Table 5. The expanded results are displayed in the table 12. We observe that an inference data volume of 100M tokens is sufficient, with larger volumes yielding only marginal improvements. Regarding the initial vocabulary size, increasing it to 150K is important to enhance performance. However, when it increases to 200K, the score shows almost no improvement, indicating that the 150K vocabulary likely already includes most of the potential final candidate tokens. Therefore, further increasing the initial vocabulary size will not bring additional benefits.

Table 12: Results for different Infer Data Volume Tokens and Initial Vocabulary Sizes.

| Infer Data Volume Tokens | 75K | 100K | 150K | 200K |
|---|---|---|---|---|
| 1M | 42.89 | 43.07 | 43.13 | 43.20 |
| 10M | 43.19 | 43.39 | 43.74 | 43.77 |
| 100M | 43.42 | 43.78 | 44.51 | 44.56 |
| 1000M | 43.45 | 43.83 | 44.53 | 44.57 |

## A.6   Details of Model Parameters

The detailed parameters of the 3 different model sizes applied in our experiment are shown in Table 13.

Table 13: Specifications of different LLMs used in the paper.

| Model Size | Layers | Model Dim | Heads | Learning Rate | Batch Size |
|---|---|---|---|---|---|
| 70M | 6 | 512 | 8 | $10.0 \times 10^{-4}$ | 32 |
| 160M | 12 | 768 | 12 | $2.5 \times 10^{-4}$ | 16 |
| 410M | 24 | 1024 | 16 | $2.5 \times 10^{-4}$ | 16 |

## A.7 Societal Impact

The adaptive tokenizer we propose is an important component of large language models. The proposed tokenizer is fine-tuned by closely monitoring the model's perplexity, enabling the language model to perform well on various tasks, such as machine translation and question answering. However, it may also face the same issues as existing subword tokenizers [19, 43], such as privacy leakage. For instance, the tokenizer might segment sensitive information, like names, addresses, or identity identifiers, into results of tokens that could be recognized in text generation. Therefore, we recommend that the training corpus undergo preprocessing to remove private information. In addition, in real-world applications, the tokenizer should be employed in conjunction with privacy-preserving technologies and specially configured filtering rules. In specific scenarios, such as medical diagnostics and legal consultations, human experts should be involved to review the tokenization results.

## A.8 List of Training Dataset

The specific corpus used for training tokenizers is from The Pile[3], and the detailed list of files is shown below.

- pile_ArXiv_025.json
- pile_ArXiv_069.json
- pile_ArXiv_070.json
- pile_ArXiv_092.json
- pile_ArXiv_098.json
- pile_ArXiv_123.json
- pile_ArXiv_124.json
- pile_ArXiv_133.json
- pile_ArXiv_134.json
- pile_ArXiv_157.json
- pile_Books3_015.json
- pile_Books3_016.json
- pile_Books3_052.json
- pile_Books3_057.json
- pile_Books3_071.json
- pile_Books3_083.json
- pile_Books3_084.json
- pile_Books3_093.json
- pile_Books3_115.json
- pile_Books3_134.json
- pile_Books3_173.json
- pile_Books3_197.json
- pile_Books3_203.json
- pile_Books3_235.json
- pile_Books3_242.json
- pile_Books3_247.json
- pile_Enron_Emails_004.json
- pile_FreeLaw_031.json
- pile_FreeLaw_083.json
- pile_FreeLaw_104.json
- pile_Gutenberg_PG-19_044.json

---

[3]URL:https://pile.eleuther.ai/

- pile_Gutenberg_PG-19_049.json
- pile_OpenSubtitles_008.json
- pile_OpenSubtitles_031.json
- pile_OpenSubtitles_037.json
- pile_OpenWebText2_011.json
- pile_OpenWebText2_050.json
- pile_OpenWebText2_063.json
- pile_OpenWebText2_108.json
- pile_OpenWebText2_118.json
- pile_OpenWebText2_132.json
- pile_OpenWebText2_157.json
- pile_OpenWebText2_162.json
- pile_OpenWebText2_212.json
- pile_OpenWebText2_216.json
- pile_OpenWebText2_242.json
- pile_OpenWebText2_245.json
- pile_OpenWebText2_256.json
- pile_Pile-CC_001.json
- pile_Pile-CC_024.json
- pile_Pile-CC_069.json
- pile_Pile-CC_076.json
- pile_Pile-CC_106.json
- pile_Pile-CC_120.json
- pile_Pile-CC_133.json
- pile_Pile-CC_181.json
- pile_Pile-CC_209.json
- pile_Pile-CC_211.json
- pile_Pile-CC_237.json
- pile_Pile-CC_254.json
- pile_Pile-CC_259.json
- pile_PubMed_Abstracts_037.json
- pile_PubMed_Abstracts_049.json
- pile_PubMed_Abstracts_054.json
- pile_PubMed_Central_028.json
- pile_PubMed_Central_053.json
- pile_PubMed_Central_067.json
- pile_PubMed_Central_069.json
- pile_PubMed_Central_085.json
- pile_PubMed_Central_123.json
- pile_PubMed_Central_125.json
- pile_PubMed_Central_132.json
- pile_PubMed_Central_149.json
- pile_PubMed_Central_165.json
- pile_PubMed_Central_173.json
- pile_PubMed_Central_215.json
- pile_PubMed_Central_220.json
- pile_Stack_Exchange_055.json
- pile_USPTO_Backgrounds_012.json
- pile_USPTO_Backgrounds_027.json
- pile_USPTO_Backgrounds_031.json
- pile_USPTO_Backgrounds_051.json
- pile_Ubuntu_IRC_001.json
- pile_Ubuntu_IRC_017.json
- pile_Ubuntu_IRC_021.json
- pile_Wikipedia_en_006.json
- pile_Wikipedia_en_009.json
- pile_Wikipedia_en_043.json
- pile_Wikipedia_en_053.json
- pile_Wikipedia_en_070.json

- pile_YoutubeSubtitles_008.json

