# OpenReview forum: "Enhancing Large Language Models through Adaptive Tokenizers"
_NeurIPS.cc/2024/Conference — NeurIPS 2024 poster_

### Official Review · Reviewer_L8Xj · 2024-07-06

**Soundness:** 2
**Presentation:** 3
**Contribution:** 3
**Rating:** 7
**Confidence:** 4

**Summary:**

The presented paper proposes an adaptive tokenization scheme that is learned jointly with an LLM and assesses its performance on downstream tasks for smaller model sizes and token budgets.

**Strengths:**

- Interesting idea to include both the loss of the current iteration as well as the loss of the previous iteration via a momentum term to stabilize the training procedure
- Interesting results on Cross-Model Adaptability of vocabularies for different model sizes

**Weaknesses:**

### Weaknesses
---


- **[medium]**: both the model size (up to 410M parameters) and the corpus size (16B tokens) seem to be on the smaller side and it is unclear if these findings would generalize to the billion level parameter level and trillion level token budget which is more representative of the scales for current state-of-the-art LLMs.
- **[medium]**: The fertility of a tokenizer is essential for comparing the average sequence length as well as performance across all supported languages i.e. a fertility of 1 would indicate that every single word is contained in the vocabulary **[1]**. I believe those values should be compared across all methods to give a better understanding.
- **[medium]**: For ARC-C, LogiQA, PIQA, and Winogrande (4/9) the scores are actually lower than the Unigram baseline or within the standard deviation range which calls the significance of the results into question. Additionally, the model size scaling experiments indicate a reduction in gains +2.29 (70M), +2.01 (160M), +1.92 (410M) over the baselines and the gap will likely continue to narrow as we scale up the model size, potentially losing its effectiveness for production-scale models. Ideally, we'd want some sort of significance tests to verify the impact.
- **[medium]**: the loss calculation in Equation 2 seems expensive for a large vocabulary $V$ as well as a large corpus. How big is the impact on the runtime and/or is this done on mini-batches? How does this scale with a bigger corpus i.e. within + beyond what is considered in Table 5? Naturally, from the ablation we would just scale both, when does this become too expensive or flatten out in terms of performance gains? Some scaling plots for both variables would be helpful for practitioners.




### Minor Comments & Typos
---

- please sort references such that [7,2] is ordered as [2,7]
- the fraction balance method in Table 5 needs brackets to reflect the correct fraction from l. 349


### References
---

- **[1]**: [How Good is Your Tokenizer? On the Monolingual Performance of Multilingual Language Models](https://aclanthology.org/2021.acl-long.243) (Rust et al., ACL-IJCNLP 2021)

**Questions:**

N/A

**Limitations:**

Might need a separate section to address limitations.

---

> ### Author Rebuttal · Authors · 2024-08-07
>
> We appreciate your positive feedback on our work and your thoughtful review.
>
> **Q1. Both the model size (up to 410M) and the corpus size (16B) seem to be on the smaller side and it is unclear if these findings would generalize to the billion level parameter level and trillion level token budget.**
>
> To demonstrate the scalability of our proposed ADAT method, we expanded our experimental results on larger models and larger corpus. Specifically, we add results(table below) from training a **1B model** on **60B corpus**. The results demonstrate that on larger models and with more data, ADAT continues to show substantial improvements over the baseline, indicating that ADAT has strong scalability.
>
> |      | Unigram | ADAT          |
> | ---- | ------- | ------------- |
> | Avg. | 49.11   | 51.20 (+2.09) |
>
> **Q2. The fertility of a tokenizer is essential for comparing the average sequence length as well as performance across all supported languages[1]. I believe those values should be compared across all methods to give a better understanding.**
>
> Regarding the fertility of tokenizers, we have included a comparative analysis of the fertility rates for all methods. We apply the four English treebanks used in Reference [1] and present the average values in the table below.
>
> |           | BPE  | BytePiece | BytePiece+ADAT | Unigram | Unigram+ADAT |
> | --------- | ---- | --------- | -------------- | ------- | ------------ |
> | Fertility | 1.25 | 1.07      | 1.08           | 1.32    | 1.49         |
>
> It's crucial to recognize that the fertility metric does not directly correlate with a model's final performance. For example, a fertility value of 1 indicates a vocabulary that covers all corpus words, similar to a bag of words model, which fails to capture intrinsic semantic connections like word roots or affixes. At the other extreme, high fertility could lead to character-level or byte-level tokenization, resulting in over-segmentation and a loss of semantic priors.
>
> [1] How Good is Your Tokenizer? On the Monolingual Performance of Multilingual Language Models (Rust et al., ACL-IJCNLP 2021)
>
> **Q3.** **The significance of the results and  losing its effectiveness for production-scale models. we'd want some sort of significance tests to verify the impact.**
>
> While ADAT scores are lower than Unigram on very few tasks, it outperforms the baseline on 7 out of 9 tasks for the 70M model and on 8 out of 9 tasks for the 410M model, with an overall improvement of approximately 2 points. This may be due to different datasets requiring different sub-optimal tokenization strategies. ADAT maintains accuracy on these datasets while improving performance on others.
>
> We used **t-tests** to compare the results of the ADAT method with the Unigram baseline to statistically validate the performance differences. For both the 70M and 410M models, the p-values comparing the ADAT method to the baseline were 0.002 and 0.0009, respectively, indicating that **ADAT achieved statistically significant performance gains over the baseline**. (p-value < 0.05).
>
> To address the concern regarding the impact of model size scaling and the potential reduction in gains, we conducted statistical tests and additional experiments as follows,
>
> - To assess the effect of the ADAT algorithm across different model sizes, we performed **t-tests** on **the performance gains provided by ADAT relative to the baseline** for the 70M and 410M models. The resulting **p-value of 0.48** indicates that the performance improvement of the ADAT method **does not significantly differ between the 70M and 410M** models, showing **consistent gains across various model sizes**.
>
> - Furthermore, As shown in the table of the response to Q1, the performance gain(2.09) is higher than that of the 410M model (1.92) and similar to that of 160M model (2.01), demonstrating the effectiveness of scalability in model size for ADAT.
>
> These results collectively confirm that the ADAT method is effective across various model sizes.
>
> **Q4. Equation 2 seems expensive. How big is the impact on the runtime and/or is this done on mini-batches? How does this scale with a bigger corpus i.e. within + beyond what is considered in Table 5?**
>
> Equation 2 is employed for generating the initial vocabulary, which is expensive for large datasets. For this, **a 5GB subset of** the corpus was used. Furthermore, during the actual calculations, an approximation of the loss is applied. The empirical runtime to produce an initial vocabulary of 150k is 914 seconds.
>
> Furthermore, we have expanded the analysis of 'infer data volume tokens' and 'initial vocabulary size'—both variables are explored within and beyond typical settings. The expanded results are displayed in the table below. We observe that an inference data volume of 100M tokens is sufficient, with larger volumes yielding only marginal improvements. Regarding the initial vocabulary size, increasing it to 150K is important to enhance performance. However, when it increases to 200K, the score shows almost no improvement, indicating that the 150K vocabulary likely already includes most of the potential final candidate tokens. Therefore, further increasing the initial vocabulary size will not bring additional benefits.
>
> |       | 75K   | 100K  | 150K  | 200K  |
> | ----- | ----- | ----- | ----- | ----- |
> | 1M    | 42.89 | 43.07 | 43.13 | 43.20 |
> | 10M   | 43.19 | 43.39 | 43.74 | 43.77 |
> | 100M  | 43.42 | 43.78 | 44.51 | 44.56 |
> | 1000M | 43.45 | 43.83 | 44.53 | 44.57 |
>
> **Q5.Typos.**
>
> Thanks for the corrections regarding the typos in sort references and Table 5. We will carefully examine the manuscript and rectify typos.
>
> **Q6. Discussion of Limitations.**
>
> Please refer to General Response.
>
> We hope we have adequately addressed your concerns. If there is still anything unclear, Please feel free to let us know during the rebuttal window.

---

> > ### Comment · Reviewer_L8Xj · 2024-08-10
> > **Response to rebuttal**
> >
> > Thanks for the additional details and experiments. I've raised my score accordingly.

---

> > > ### Author Response · Authors · 2024-08-12
> > > **Thank you**
> > >
> > > Thank you for your recognition and for raising your score! Your support is greatly appreciated.

---

### Official Review · Reviewer_GBKH · 2024-07-12

**Soundness:** 2
**Presentation:** 3
**Contribution:** 2
**Rating:** 5
**Confidence:** 2

**Summary:**

This study introduces an adaptive tokenizer whose development is integrated with the performance of the LLM. The tokenizer has the particularity that it is fine-tuned based on the model’s perplexity during training. Empirical results show that this approach improves accuracy compared to “traditional” tokenization methods.

**Strengths:**

- Well motivate and easy to understand
- Experiments are very comprehensive
- Insightful ablation study
- Answer most of the questions that one may have for a tokenizer: impact on perplexity, accuracy in downstream tasks, and performance depending on the vocabulary size,

**Weaknesses:**

- There are only comparisons with very common tokenizers: BPE, BytePiece, and Unigram. This is enough to guide engineers but for scientific work, I would expect comparisons with related work, even if they are not widely adopted. For instance, how does it compare to other adaptive tokenizers such as the one proposed by “Task-Adaptive Tokenization: Enhancing Long-Form Text Generation Efficacy in Mental Health and Beyond” (not cited)?
- The limitations of this work are not very well discussed. When does ADAT completely fail? For which scenario shouldn’t we use it?

**Questions:**

The main suggestion that I have for this work would be to compare it with other tokenizers that might not be mainstream but that have been shown to perform better. Comparing it with other Task-Adaptive Tokenizers would be a start.

**Limitations:**

See weaknesses.

---

> ### Author Rebuttal · Authors · 2024-08-07
>
> Thank you for your efforts to enhance the quality of our manuscript. We appreciate the issues you identified, and we believe we have thoroughly clarified and addressed them as follows.
>
> **Q1. There are only comparisons with very common tokenizers: BPE, BytePiece, and Unigram. This is enough to guide engineers but for scientific work, I would expect comparisons with related work, even if they are not widely adopted. For instance, how does it compare to other adaptive tokenizers such as the one proposed by “Task-Adaptive Tokenization: Enhancing Long-Form Text Generation Efficacy in Mental Health and Beyond” (not cited)?**
>
> Thanks for your feedback regarding the comparison method in our experiments. Task-adaptive tokenizers represent a promising direction, yet they **serve a different purpose from the methodology outlined in our paper**. Like BPE and Unigram tokenizers, our proposed model ADAT is designed to learn a general tokenizer. In contrast, the task-adaptive tokenizer[1] you mentioned focuses on integrating tokenizers derived from varying data distributions.
>
> In the current experimental setup, we incorporated the task-adaptive tokenizer[1] by utilizing data from 'the pile' as Distribution A, while employing the training dataset from test set ARC-E, PIQA and SciQ as Distribution B. This approach facilitated the generation of a vocabulary through task-adaptive tokenization. The outcomes, presented in the table below, illustrate that the task-adaptive method exhibits robust performance on tasks ARC-E, PIQA and SciQ. However, on the other 6 tasks, its performance is inferior to ADAT, and on 5 of those tasks, it falls below the baseline, attributable to its merge strategy that favours task-specific tokens, consequently neglecting universal tokens.
>
> Given that the task-adaptive methodology accesses an expanded dataset from Distribution B (the training set of the test set ARC-E, PIQA and SciQ), the findings suggest its limitations as a universally applicable tokenization strategy. Detailed discussions and comparisons with Reference[1] are planned for the final version of our paper to thoroughly investigate these observations.
>
> These findings are significant as they demonstrate the robustness of our method, even when compared to more specialized, task-adaptive tokenization strategies. We hope this additional analysis addresses your concerns and further validates the effectiveness of our proposed method.
>
> |                         | ARC-C | ARC-E | Boolq | Lambda | LogiQA | PIQA  | SciQ  | SST-2 | Winogrande | Avg.  |
> | ----------------------- | ----- | ----- | ----- | ------ | ------ | ----- | ----- | ----- | ---------- | ----- |
> | Unigram                 | 19.54 | 37.04 | 53.06 | 17.27  | 23.20  | 60.50 | 68.10 | 49.77 | 51.46      | 42.22 |
> | ADAT(Ours)              | 18.46 | 40.57 | 61.19 | 17.97  | 24.22  | 59.93 | 72.40 | 54.24 | 51.62      | 44.51 |
> | Task-specific tokenizer | 17.15 | 37.82 | 55.63 | 14.36  | 22.73  | 59.58 | 69.20 | 49.08 | 50.36      | 41.77 |
>
> [1] Task-Adaptive Tokenization: Enhancing Long-Form Text Generation Efficacy in Mental Health and Beyond. EMNLP2023
>
> **Q2. Discussion of Limitations.**
>
> We apologize for not clearly and separately discussing the limitations of our method in the manuscript. We will add the following discussion of limitations to the beginning of the Conclusion section as a separate part in the final version.
>
> The adaptive nature of our proposed tokenizer method introduces variations in tokenizers across different model sizes, leading to inconsistent vocabularies. This inconsistency complicates tasks such as knowledge distillation and speculative decoding, which rely on the assumption of a uniform vocabulary across both smaller and larger models.
>
>
>
> We hope that our answer has addressed your concerns. Please feel free to let us know during the rebuttal window if there is still anything unclear. We appreciate your suggestions and comments! Thank you!

---

> ### Author Response · Authors · 2024-08-10
>
> Dear Reviewer GBKH,
>
> Thank you once again for the time you spent reviewing our paper and for your efforts to enhance the quality of our manuscript. We hope that our response can fully address your concerns.
>
> Given that the discussion period concludes on August 13th, we would appreciate any further questions you might have about our paper,  and we are glad to have a discussion with you over the coming days. If all the concerns you have raised have been addressed by our response, would you mind considering re-evaluating our work based on our response?

---

> > ### Comment · Reviewer_GBKH · 2024-08-12
> >
> > Thank you for the additional experiments.
> > I increased my score but also significantly decreased my confidence score. I was quite sure that your work was not the first to propose an adaptive tokenizer for this use case.
> > Especially in neural machine translation, adaptive tokenization has been studied a lot. But I have to admit that I can't find the papers that I'm thinking of, hence my late reply.

---

> > > ### Author Response · Authors · 2024-08-13
> > >
> > > We truly appreciate your revised evaluation and the increase in score, your support is valuable to our work. Thank you very much.
> > >
> > > We have conducted further research on the existing work of adaptive tokenizers, including in the field of neural machine translation (NMT). For example,
> > >
> > > ONE-Piece[1] proposed a subword tokenization using morphological segmentation and vocabulary communication to address OOV problem. This method leverages the correspondence between two languages to create the tokenizer, specifically tailored for translation tasks. Moreover, it differs from our approach by not adapting to the model. AT[2] adapts the tokenizer of the pretrained model to transfer the pretrained language model to new domains. It defines adaptive tokenization as augmenting the existing tokenizer and fixed subword embeddings with new entries from a novel corpus. But it does not create a general tokenizer. Our proposed method, which aims to build a general tokenizer, differs in purpose from the mentioned approaches. Considering the differences in setting—such as ONE-Piece[1] being designed for NMT tasks that use two corresponding languages to generate a tokenizer, and AT[2] focusing on expanding an existing vocabulary to new domains—these methods are not directly comparable to ours.
> > >
> > > We will further discuss the mentioned references in the final version.
> > >
> > > Thank you once again for your time and efforts in reviewing our paper and for raising your score.
> > >
> > >
> > >
> > > [1] Should we find another model?: Improving Neural Machine Translation Performance with ONE-Piece Tokenization Method without Model Modification. (Park et al., NAACL 2021)
> > >
> > > [2] Efficient domain adaptation of language models via adaptive tokenization[J]. arXiv preprint arXiv:2109.07460, 2021.

---

### Official Review · Reviewer_aWgE · 2024-07-15

**Soundness:** 3
**Presentation:** 4
**Contribution:** 3
**Rating:** 7
**Confidence:** 4

**Summary:**

This paper proposes a method to learn the tokenizer of a language model as part of language model training. The method works by combining a compression loss (which is also used by classical tokenization algorithms) with a language modeling loss, and iteratively removing tokens that contribute the least to the combined loss. The authors show that this method leads to improved performance on several downstream tasks compared to commonly-used tokenizers and also conduct an extensive quantitative analysis of various design choices (e.g., vocabulary size, corpus size, initial vocabulary size).

**Strengths:**

The authors take a fresh look at tokenization and come up with a novel method that is simpler than many alternative approaches while still resulting in clear performance improvements. The presentation of the method is clear and easy to follow. The experiments are quite extensive and include analyses into the different components of the method. The results show systematic performance gains compared to the standard tokenizers. Overall, I think that this is a valuable contribution to the emerging field of tokenization research.

**Weaknesses:**

There are a few places where decisions of the authors seemed questionable to me and/or where I would have liked to see more details:

- The authors make the assumption that an individual token $x$ contributes to the overall language modeling loss only via the cross-entropy loss in places where $x$ is the to-be-predicted token. However, this ignores the impact that $x$ has on the language modeling loss as part of the left context (i.e., when $x$ is among the tokens processed before predicting the next token). I understand that this might be a necessary simplification to make the method feasible, but I would have expected a more in-depth discussion of this limitation and its potential repercussions.
- Based on the details provided in the paper, the setup of evaluating performance in terms of perplexity seems to be flawed: if I understand correctly, the adaptive tokenizer is trained using the token losses on the Pile, and the Pile is then again used to evaluate the different methods in terms of perplexity. I think this gives an unfair advantage to the adaptive method. The authors should either (i) evaluate perplexity on a different dataset not used for training or (ii) refrain from using perplexity as an evaluation measure.
- I would have liked to see a more in-depth analysis of the compute costs of the adaptive tokenizer compared to the standard tokenizers.

**Questions:**

These are mostly comments and smaller points:

- Citation [8] should be NeurIPS 2023, not 2024.
- An important paper not cited is [Nawrot et al. (2023)](https://aclanthology.org/2023.acl-long.353/).
- 84: "Neurall" -> "Neural"
- 86: "Stringent" -> "stringent"
- Figure 1: "Troditional" -> "Traditional"
- 194: "We" -> "we"
- 226: How did you test for significance? Please mention the statistical test and exact results or else rephrase (e.g., "substantial").
- 195: How exactly do you conduct these evaluations? Are they zero-shot? Please provide more details.
- 235: "examines" -> "examine"
- 244: "trianing" -> "training"
- 4.6.2: Why did you not test above 150K tokens?

**Limitations:**

The authors say that limitations are discussed in the "Experimental Setup" subsection, but I could not find any discussion of the limitations there.

---

> ### Author Rebuttal · Authors · 2024-08-07
>
> We appreciate your positive feedback on our work and your insightful comments.
>
> **Q1. This ignores the impact that token $x$ has on the language modeling loss as part of the left context. I would have expected a more in-depth discussion.**
>
> Thanks for your insightful feedback. In response to your comments, we conducted additional experiments to examine the impact of token $x$ when it appears as part of the left context in the prediction target. We allocated the predicted token's loss to its left-hand side tokens based on attention weights. To manage computational costs effectively, we implemented a lookback window that only considers the immediate token to the left for loss updates.
>
> These experiments were carried out on a model with 70M parameters, and the results are detailed in the table below. The findings indicate that incorporating this "lookback loss" has a minimal impact on model performance. This suggests that while the contextual influence of $x$ is indeed present, its effect is sufficiently captured under our current modeling framework without necessitating significant computational overhead.
>
> |  | Unigram | ADAT | ADAT+Win |
> | ----- | ---- | ------ | ------ |
> | Avg. | 42.22 | 44.51 | 44.38 |
>
> **Q2. PPL evaluation method may unfairly advantage adaptive tokenizer.**
>
> Thank you for your valuable comments regarding our evaluation methodology. I would like to provide clarifications on two key points you raised:
>
> Firstly, as stated in line 193 of our manuscript, the perplexity (ppl) evaluation metric was conducted on the  **PG19 dataset, not the Pile dataset**. This distinction ensures that our training and testing were performed on separate datasets, effectively mitigating any potential issues of data leakage.
>
> Secondly, to comprehensively assess the effectiveness of our method, we reported the accuracy (ACC) of our model across nine different datasets. These results robustly validate the performance of our approach.
>
> In the final version of our paper, we will clarify these evaluation details more explicitly to prevent any possible misunderstandings.
>
> **Q3. I would have liked to see a more in-depth analysis of the compute costs.**
>
> In the table below, we outline the computational costs for standard tokenizers, the proposed ADAT, and the training phases of an LLM. Although ADAT introduces additional computational expenses, these costs are minimal and marginal compared to the significant resources required for LLM training.
>
> We analyzed the empirical runtime introduced by ADAT. The tokenizer optimization involves 5 epochs, where each epoch consists of training the LLM on a 0.3B corpus, followed by inference on a 0.1B corpus, and concludes with a vocabulary pruning step (90 seconds for a 100K tokens vocabulary). Therefore, the total computational cost of the tokenizer optimization process is calculated as,
> $$5 \times (0.3B \text{ training} + 0.1B \text{ inference}+\text{ pruning time }) = 1.5B \text{ training} + 0.5B \text{ inference} + 450s$$
>
> ADAT introduces an additional training cost of 1.5B tokens and an inference cost of 0.5B tokens, along with minimal vocabulary pruning time. Compared to the hundreds of billions or even trillions of tokens required for LLM training, these computational costs are negligible.
>
> To measure runtime, we used 8 NVIDIA A100 GPUs, an Intel 8378A CPU, and PyTorch 2.1.2 with CUDA 12.1. As illustrated in the table below, the Unigram tokenizer consumes 0.33 CPU hours, whereas ADAT requires around 2 GPU hours. The LLM training phase demands significantly more resources, exceeding 500 GPU hours [1]. This comparison underscores that the additional computational expense introduced by ADAT is relatively insignificant, further justifying its use given its potential benefits.
>
> |                |Unigram| ADAT-70M|Pythia-70M|
> | ------------ | ------ | -------- | ---------- |
> | Runtime | 0.33 CPUh | 2 GPUh  | 510 GPUh   |
>
> [1] Pythia: A Suite for Analyzing Large Language Models Across Training and Scaling. S Biderman.
>
> **Q4. References.**
>
> We will include and discuss the reference [2] in the final version.
>
> [2] Nawrot P, Chorowski J, Lancucki A, et al. Efficient Transformers with Dynamic Token Pooling
>
> **Q5. Lin226. How did you test for significance? Please mention the statistical test and exact results or else rephrase (e.g., "substantial").**
>
> Sorry for the misleading. We use the word "significant" merely to describe the degree of improvement. We will replace it with "substantial" or "considerable."
>
> **Q6. How exactly do you conduct these evaluations?**
>
> We apply five-shot evaluations in experiments, we will explicitly clarify this in the final version.
>
> **Q7. 4.6.2 Why did you not test above 150K tokens.?**
>
> As shown in Table 2 of the manuscript, increasing the initial vocabulary size up to 150K markedly enhances performance. However, when it increases to **200K**, the score is **44.56**, which is comparable to the score of 44.51 for 150K, indicating that the 150K vocabulary likely already includes most of the potential final candidate tokens. Therefore, further increasing the initial vocabulary size will not bring additional benefits.
>
> **Q8 Typos.**
>
> Thanks for your suggestions and corrections. We will correct all typos and revise the year information regarding citation [8] in the manuscript.
>
> **Q9. Discussion of Limitations.**
>
> Please refer to General Response.
>
> If there is still anything unclear, Please feel free to let us know during the rebuttal window.

---

> > ### Comment · Reviewer_aWgE · 2024-08-13
> > **Response to the Authors' Rebuttal**
> >
> > I thank the authors for these helpful explanations. I have raised my score.

---

> > > ### Author Response · Authors · 2024-08-14
> > > **Thank you**
> > >
> > > We are grateful for your recognition and for increasing your score. Thank you so much!

---

### Official Review · Reviewer_t7me · 2024-07-16

**Soundness:** 3
**Presentation:** 2
**Contribution:** 3
**Rating:** 6
**Confidence:** 4

**Summary:**

Language model vocabularies are typically learned using frequency-based objectives. This objective does not entirely align with the language modeling objective–the task for which these vocabularies are ultimately used–and may therefore cause a bottleneck in language model performance. This proposes a new tokenization method that addresses this issue. Specifically, the vocabulary is optimized using an objective that incorporates both frequency-based and language-modeling losses. The method is relatively simple: use the Unigram LM tokenization method albeit with an altered loss function (i.e., not just unigram negative log-likelihood but also the cross entropy under a language model trained using the vocabulary under consideration). The two loss terms can be balanced using different functional forms and loss terms from previous iterations can be incorporated. Empirically, the authors find that their approach leads to better model performance across a variety of tasks. They perform ablations on various design choices.

**Strengths:**

* This work takes a step towards the more end-to-end learning of language models, exploring perhaps one of the last remaining components in language modeling that has not been well optimized yet.
* The proposed method is a simple extension of a widely used algorithm, which could make its adoption easier.
* Empirical results show how sensitive models are to tokenization, which is something that is perhaps underestimated and is an important point that the community should be aware of.
* Even though the method is rather computationally expensive, it appears that using smaller models in the computation of the CE component of the tokenization loss is also effective. Training these smaller LMs can be considered a drop in the bucket compared to the total amount of computation used for training larger language models, although its unclear from the experiments how much smaller the tokenization LM can be compared to the final LM while still leading to good performance

**Weaknesses:**

* Ultimately, the algorithm may not be practically feasible, given that it requires training an LM at each iteration of vocabulary pruning. This is a very important aspect that the authors do not discuss, either in terms of runtime analysis or empirical runtimes.
* The writing is fairly poor and imprecise. Comments such as "maintaining the stability of the vocabulary is crucial to ensure consistent learning outcomes" (line 159) are very vague. What is the "stability of the vocabulary"? What does it mean to have "consistent learning outcomes"? There are numerous other examples such as this and collectively, they will leave readers confused or perhaps even misinformed.
* Stylistic points: There is a lot of redundancy between the introduction and section 2.1 that can be eliminated.
* There are a huge number of spelling errors. Please run the manuscript through a spell checker!

**Questions:**

* In equation 1, the notation suggests that you’re optimizing the dataset. I’m guessing this is not the case though. Could you clarify?
* Was switching between V and D to denote the vocabulary and D and T to denote the dataset an intentional choice?
* How much more computation does ADAT take? Unclear… computational complexity should be discussed
* Do you have Insights about why 410M vocab doesnt lead to big improvements for the smaller model?
* The jump from 70M to 410M parameters isn’t huge. Do you think a 70M model could be used in ADAT for finding the vocabulary of a much larger model? This is perhaps the only way I see the algorithm being scalable.
* Smaller points: The relationship between inference speed and sequence length (discussed first in section 3) should be made explicit; I believe lines 121-22 are a misstatement. Should either be least -> most or increasing -> decreasing

**Limitations:**

Limitations are generally not discussed. The most important points that the authors need to address are the changes to the computational complexity of the tokenization step and the fact that only English is explored empirically.

---

> ### Author Rebuttal · Authors · 2024-08-07
>
> Thank you for your careful reading of our paper and valuable comments.
>
> **Q1. The algorithm's repeated LM training and unaddressed runtime issues may limit practicality.**
>
> Our algorithm is feasible in practice, because the optimization phase of the tokenizer only requires a minimal amount of data (0.3B), compared to the full training of the Large Language Model (LLM), resulting in only slight computational overhead.
>
> We analyzed the empirical runtime introduced by ADAT. To measure runtime, we used 8 NVIDIA A100 GPUs, an Intel 8378A CPU, and PyTorch 2.1.2 with CUDA 12.1. The tokenizer optimization involves 5 epochs, where each epoch consists of training the LLM on a 0.3B corpus, followed by inference on a 0.1B corpus, and concludes with a vocabulary pruning step (90 seconds for a 100K tokens vocabulary). Therefore, the total computational cost of the tokenizer optimization process is calculated as,
> $$5 \times (0.3B \text{ training} + 0.1B \text{ inference}+\text{ pruning time }) = 1.5B \text{ training} + 0.5B \text{ inference} + 450s$$
>
> In contrast, the full-scale training of the LLM incurs significantly higher computational costs. For instance, when training models with a 16B and 60B corpus, the tokenizer optimization accounts for only 4.17% and 1.04% of the total training time, respectively. The Pythia-70M model takes 510 GPU hours to train with the full Pile dataset [1], and exceeds the tokenizer optimization's computational cost by over **255 times**. Therefore, the additional computational cost introduced by our method is minimal, making it feasible in practice.
>
> |         | Tokenizer Optimization | Training on 16B | Training on 60B | Pythia Report |
> | ------- | ---------------------- | --------------- | --------------- | ------------- |
> | RunTime | 2 GPUh                 | 48 GPUh         | 192 GPUh        | 510 GPUh      |
>
> [1] Pythia: A Suite for Analyzing Large Language Models Across Training and Scaling. S Biderman.
>
> **Q2. Writing lacks clarity.**
>
> We appreciate your feedback regarding the readability of our paper. In the revised manuscript, we will ensure more thorough proofreading and polishing to improve the overall quality and precision of the writing. Specifically, we clarify as follows,
>
> \- **Vocabulary stability** refers to the consistency of each token's score relative to the previous round during each iteration of vocabulary pruning. To mitigate significant fluctuations, we introduced a momentum mechanism to smooth these changes and ensure score stability.
> \- **Consistent Learning Outcomes** refers to achieving stable and reliable resulting vocabulary with same data and strategies, ensuring that the outcomes do not significantly vary due to different conditions or random factors.
>
> **Q3. Redundancy between the introduction and section 2.1.**
>
> Thank you for your suggestions. We will revise Section 2.1 of the related work to eliminate redundancy.
>
> **Q4. Why 410M vocab doesnt lead to big improvements for the smaller model?**
>
> This phenomenon may be due to larger models with more parameters are better equipped to capture complex token relationships. Thus, vocabularies from larger models have higher model capabilities, leading to smaller performance improvements when applied to smaller models. In contrast, vocabularies generated by smaller models are less dependent on extensive model capabilities, and thus, when applied to larger models, they are able to sustain better performance gains.
>
> **Q5. Do you think a 70M model could be used in ADAT for finding the vocabulary of a much larger model?**
>
> Thank you for this insightful question. Yes, I believe it is a feasible way to enhance scalability.  We add experimental results of 70M-ADAT on a larger model (1B) in the table below. As the table shows, the 70M-ADAT also improves performance compared to Unigram. This demonstrates that the vocabulary found by 70M-ADAT exhibits good generalizability across different model sizes.
>
> Based on our analysis of compute costs in response to Q1, directly scaling ADAT to a large model is within an acceptable cost range compared to training a large model.
>
> | 1B model with Unigram | 1B model with 70M-ADAT |
> | --------------------- | ---------------------- |
> | 48.63                 | 49.68                  |
>
> **Q6. Only English is explored empirically.**
>
> To verify the effectiveness of ADAT in other languages, we further assess our method using a **mixed corpus of Chinese and English**. We evaluate the performance on the same English benchmarks as in the manuscript and the Language Model Evaluation Opencompass for the Chinese benchmark FewCLUE [2]. The results shown in the table below indicate that ADAT shows a 2.13 increase on the Chinese benchmark and a 2.28 for English benchmarks, demonstrating its effectiveness in other languages.
>
> | benchmark | English | Chinese |
> | --------- | ------- | ------- |
> | Unigram   | 37.75   | 44.75   |
> | ADAT      | 40.03   | 46.88   |
>
> [2] FewCLUE: A Chinese Few-shot Learning Evaluation Benchmark
>
> **Q7.  Smaller points and misstatements.**
>
> We appreciate your attention to detail, and we will carefully examine the manuscript and correct all spelling errors and misstatements.
>
> - Equation 1:  The vocabulary $V$ will be optimized using Equaton 1, which will be rewritten as $ \min_V \textbf{Length}(D_o,V)- \lambda \textbf{Acc}(D,M,V) , \vert V\vert \leq N $ in the final version.
>
> - Notation of vocabulary and dataset:  We will use consistent notation.
>
> - Inference Speed and Sequence Length: For a given text, longer token sequences extend inference time and reduce inference speed due to increased computational demands. We will include clearer expressions in the final version.
>
> - Line 121: We will change "least" to "most".
>
> **Q8. Discussion of Limitations.**
>
> Please refer to General Response.
>
> We hope we have adequately addressed your concerns. If there is still anything unclear, please feel free to let us know during the rebuttal window.

---

> > ### Comment · Reviewer_t7me · 2024-08-11
> >
> > Thank you for the clarifications. I am raising my score slightly

---

> > > ### Author Response · Authors · 2024-08-12
> > > **Thank you**
> > >
> > > We sincerely appreciate your recognition and thank you very much for raising your score!

---

### Author Rebuttal · Authors · 2024-08-07

We sincerely appreciate the reviewers' time and the valuable feedback they have provided for our paper. These constructive comments have been instrumental in enhancing the quality of our work. Here is one common concern we would like to address:

**General Response: Discussion of Limitations.**

We apologize for not clearly and separately discussing the limitations of our method in the manuscript. We will add the following discussion of limitations to the beginning of the Conclusion section as a separate part in the final version.

The adaptive nature of our proposed tokenizer method introduces variations in tokenizers across different model sizes, leading to inconsistent vocabularies. This inconsistency complicates tasks such as knowledge distillation and speculative decoding, which rely on the assumption of a uniform vocabulary across both smaller and larger models.

---

> ### Author Response · Authors · 2024-08-10
>
> Dear Reviewers,
>
> Many thanks to all reviewers for your insightful and valuable feedback. We hope we have addressed your concerns. Please let us know if you have any remaining concerns and questions.

---

### Decision · Program_Chairs · 2024-09-25

**Decision:**

Accept (poster)

**Comment:**

This paper presents an interesting study of how different tokenizers can affect the quality of an LLM. This is a less studied problem in existing literature, and the authors do a good job at motivating the problem and providing analyses of results. The authors have also presented numbers that show that their method can work not only for smaller models nut also of models upto 1B parameters in size. The paper can benefit from better writing and from expanded set of experiments as noted by the reviewers.